# Beyond Machine Interpretation: Learning from Expert Over-Reads Improves ECG Diagnosis

**Sunwoo Kwak**[1]                                    SK3355@CORNELL.EDU

**Fengbei Liu**[1]                                    FL453@CORNELL.EDU

**Nusrat B. Nizam**[1]                                NN284@CORNELL.EDU

**Ilan Richter**[2]                                   IR2498@CUMC.COLUMBIA.EDU

**Nir Uriel**[2]                                      NU2126@CUMC.COLUMBIA.EDU

**Peter M. Okin**[3]                                  POKIN@MED.CORNELL.EDU

**Mert R. Sabuncu**[1,4,5]                            MSABUNCU@CORNELL.EDU

[1] *Cornell Tech, New York, NY, USA*

[2] *Department of Medicine, Columbia University Irving Medical Center, New York, NY, USA*

[3] *Department of Medicine, Weill Cornell Medicine, New York, NY, USA*

[4] *Department of Radiology, Weill Cornell Medicine, New York, NY, USA*

[5] *School of Electrical and Computer Engineering, Cornell University, Ithaca, NY, USA*

**Editors:** Accepted for publication at MIDL 2026

## Abstract

Automated machine-read ECG interpretations are widely used in clinical practice but often unreliable, leading to systematic diagnostic errors. This work investigates how training with cardiologist over-reads impacts model accuracy and clinical reliability. Using a large paired corpus of over two million ECGs containing both machine and expert interpretations, we evaluate three learning paradigms: (i) supervised learning on expert over-read labels, (ii) Self-training that extends expert supervision to public ECGs, and (iii) multimodal contrastive learning with CLIP and NegCLIP. Across all settings, models trained with expert over-read data consistently outperform those trained on machine-read labels, especially for rare but clinically important conditions. Self-training and NegCLIP further demonstrate scalable strategies to propagate expert knowledge beyond labeled datasets. These findings highlight the essential role of expert over-reads in developing trustworthy and clinically aligned ECG AI systems. Code and model can be found at this link : https://github.com/tyoung089/Overread_ecg

**Keywords:** ECG, Expert Over-read, Self-training, Contrastive learning, Clinical AI

## 1. Introduction

Electrocardiography (ECG) is a cornerstone of early cardiovascular diagnosis, yet expert interpretation requires substantial specialty training, and the clinical workforce capable of performing high-quality ECG over-reads is shrinking—nearly half of U.S. counties have no cardiologist specializing in ECG interpretation (Kim et al., 2024). Consequently, many hospitals and clinics now rely heavily on automated machine-read interpretations produced by device-embedded algorithms (Smulyan, 2019; Cook et al., 2020). However, these systems are not reliably accurate. Multiple studies report misclassification rates of 20–35% in arrhythmias, conduction abnormalities, and ischemic patterns (Schläpfer and Wellens, 2017; Kraik

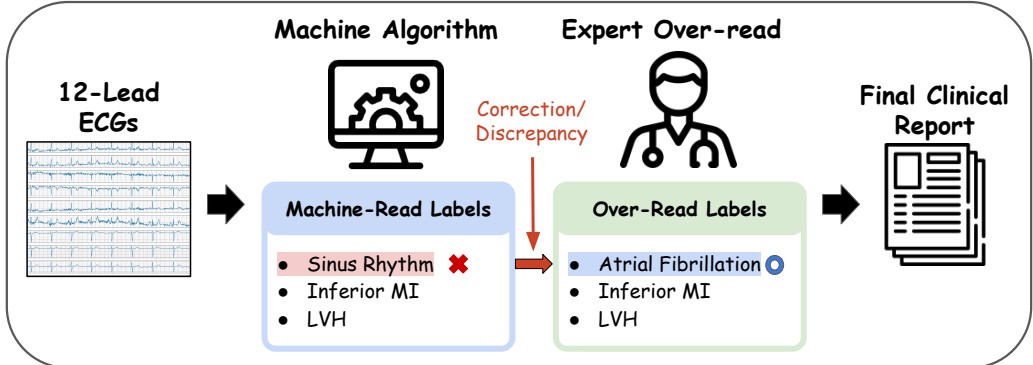

Figure 1: **Workflow of ECG interpretation and labeling.** 12-lead ECGs are first acquired during routine clinical care and automatically interpreted by vendor-provided machine algorithms. These machine-read diagnoses are subsequently reviewed and corrected by board-certified cardiologists through a standardized over-reading process, producing the final clinical report. This sequential pairing of machine-read and expert over-read interpretations enables direct analysis of diagnostic discrepancies and supports model training using expert-verified labels.

et al., 2025), prompting clinical guidelines to recommend that all automated ECG outputs be verified by cardiologists' over-read before informing patient care (Heidenreich et al., 2021). Despite this, both routine practice and much of ECG machine learning research continue to depend on machine-read labels, primarily because large-scale expert-annotated corpora remain difficult and costly to obtain (Hong et al., 2020). Influential public datasets, such as MIMIC-IV-ECG (Gow et al., 2023) and the PhysioNet 2021 Challenge (Reyna et al., 2021) –have accelerated methodological progress but rely largely on automated or partially verified interpretations. Models trained on such supervision risk inheriting systematic errors embedded in machine-read diagnostics, especially for subtle or low-prevalence conditions, leaving a persistent gap between algorithmic performance and clinical reliability.

To address this challenge, we curated a large corpus of over two million 12-lead ECGs, each containing both the original machine-read interpretation and a cardiologist over-read provided by experts with more than 30 years of experience. The sequential workflow to acquire machine-read and over-read labels is described in Figure 1. This paired design enables the direct measurement of diagnostic discrepancies between machine and expert readings and allows models to learn from validated expert judgments rather than noisy automated labels. Our central hypothesis is that models trained on over-read data achieve higher diagnostic accuracy, robustness, and clinical alignment than those trained solely on machine-read interpretations.

In our study, the first two approaches—supervised and self-training—require label extraction from raw over-read text descriptions, while the contrastive approach can operate directly on the unstructured text itself. We evaluate three paradigms. **(i) Supervised learning:** a multi-label ResNet-50 classifier (He et al., 2016) trained on expert over-read labels, serving as a direct benchmark for expert-informed ECG interpretation. **(ii) Self-**

**training:** a FixMatch-style framework (Sohn et al., 2020) that extends the supervised model by treating large public ECG datasets with machine-read labels (e.g., MIMIC-IV-ECG) as unlabeled samples, effectively leveraging expert supervision without requiring additional annotation. **(iii) CLIP-style contrastive learning:** a multimodal contrastive objective inspired by CLIP (Radford et al., 2021) and extended with NegCLIP ideas (Yuksekgonul et al., 2023), which jointly trains ECG signals with raw expert over-read text to align signal–text representations and explicitly separate machine biases from expert-level semantics.

Across these paradigms, each approach offers different strengths. Supervised learning provides a strong and interpretable baseline but is limited to the extracted label set. Self-training improves performance by combining expert-labeled and large public datasets, though it requires longer training and substantial unlabeled corpora. CLIP-style contrastive learning achieves comparable accuracy while remaining highly scalable, enabling new diagnostic phrases or expert text to be incorporated without full retraining. Crucially, in all settings, models trained on expert over-read labels consistently outperform those trained on machine-read data. While supervised learning and self-training already benefit from expert over-read supervision, the remaining gap between CLIP and NegCLIP highlights the value of paired machine-read and over-read labels. By explicitly contrasting machine vs. expert interpretations, NegCLIP better suppresses vendor-specific biases and yields higher-precision automatic diagnosis models. This performance gap illustrates the clinical risk of relying solely on machine-read interpretations and highlights the necessity of large-scale expert over-read data for developing reliable ECG AI systems.

**Key contributions:** (1) We curate a large-scale paired ECG corpus—over two million 12-lead recordings with both machine-read interpretations and expert cardiologist over-reads—enabling direct quantification of diagnostic discrepancies at scale. (2) Using this corpus, we systematically evaluate supervised, self-training, and CLIP-style contrastive frameworks to assess how expert over-read supervision impacts model accuracy, robustness, and clinical alignment. (3) We show that expert-supervised models consistently outperform those trained on machine-read labels across major diagnostic categories.

## 2. Related Work

Machine-read ECGs are widely used in clinical workflows, yet clinically important inaccuracies persist; authoritative reviews consistently recommend that automated outputs be verified by clinician over-reads before informing care (Schläpfer and Wellens, 2017; Smulyan, 2019; Heidenreich et al., 2021). Despite this, large-scale expert annotations remain difficult to obtain, leading most public corpora to rely on partially verified or machine-generated interpretations.

Open ECG datasets such as MIMIC–IV–ECG and the PhysioNet 2021 Challenge have accelerated progress, but their label provenance is heterogeneous and often dominated by device-generated or mixed pipelines rather than comprehensive expert over-reads at scale (Gow et al., 2023; Reyna et al., 2021). Models trained solely on such labels risk propagating systematic noise, especially for subtle, low-prevalence, or clinically consequential conditions (Hong et al., 2020). Only a few prior studies have examined machine–expert discrepancies, and none provide large paired corpora enabling systematic quantification across millions of ECGs.

In parallel, ECG foundation models emphasize scale and transfer. Recent efforts such as ECG-FM and ECGFounder pretrain on over one to ten million ECGs and report strong performance across diagnostic and prognostic tasks (McKeen et al., 2025; Li et al., 2025). However, these efforts also rely primarily on machine-read or heterogeneous labels, reinforcing the field's dependence on imperfect supervision.

Our work complements these directions by focusing on who provides the supervision. Using a uniquely paired corpus containing both machine-read interpretations and cardiologist over-reads, we quantify how expert-verified supervision alters downstream reliability relative to machine-read labels. We further explore scalable learning paradigms—supervised learning, FixMatch-style self-training (Sohn et al., 2020), and multimodal contrastive learning (Radford et al., 2021; Yuksekgonul et al., 2023)—to propagate expert signal and align model behavior with validated clinical judgment.

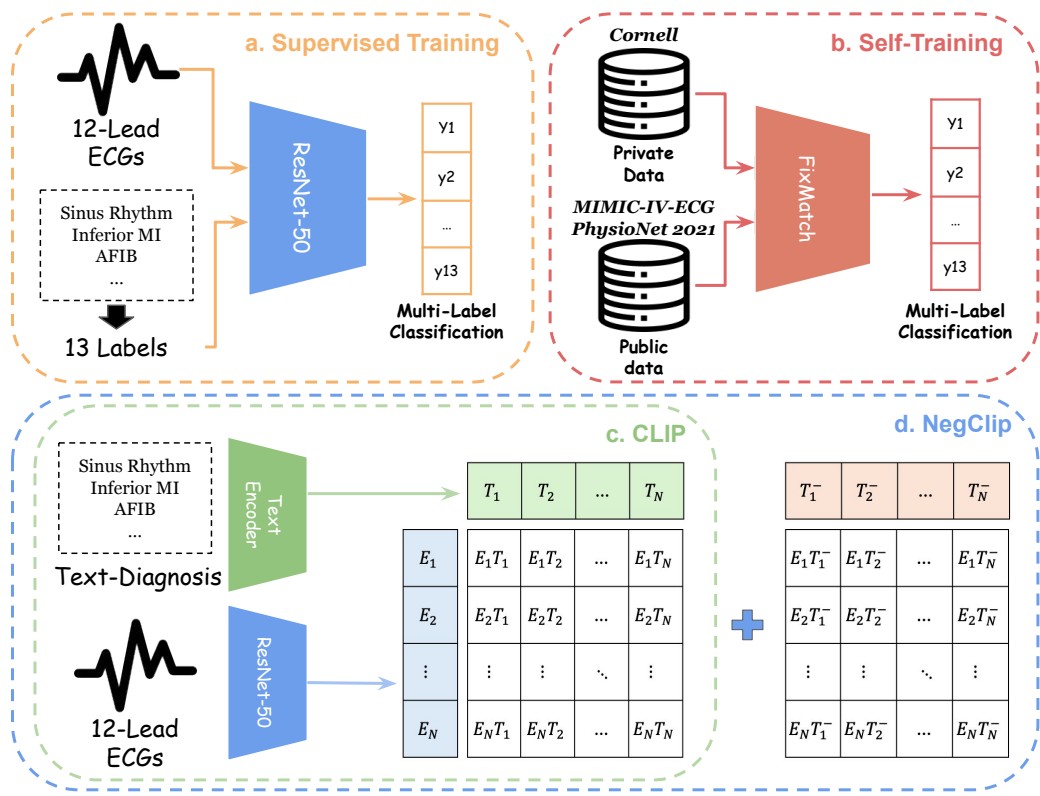

Figure 2: Overview of the four learning paradigms evaluated in this study: (a) **Supervised learning** using expert over-read labels, (b) **Self-training** that leverages unlabeled public ECGs through pseudo-labeling, (c) **CLIP-style contrastive learning** aligning ECG signals with expert text, and (d) **NegCLIP** extending CLIP with hard-negative sampling for finer signal–text discrimination. Together, these frameworks represent complementary strategies for integrating expert knowledge into ECG AI models.

## 3. Methods

**Preliminary** We denote the over-read available training set as $\mathcal{D}_L = \{\mathbf{x}_i, \mathbf{t}_i, \hat{\mathbf{t}}_i\}_{i=1}^{|\mathcal{D}_L|}$, where $\mathbf{x}_i \in \mathcal{X} \subset \mathbb{R}^{L \times S}$ is the ECG signal with shape $L$ leads and $S$ length, and $\mathbf{t}_i$ and $\hat{\mathbf{t}}_i$ are the corresponding over-read and machine-read textual reports. We further derive over-read diagnostic labels $\mathbf{y}_i \in \{0,1\}^C$ from $\mathbf{t}_i$ and machine-read labels $\hat{\mathbf{y}}_i \in \{0,1\}^C$ from $\hat{\mathbf{t}}_i$ using the same label extraction process, $C$ is the number of classes and both labels are multi-hot vectors. We also define public dataset with no over-read as $\mathcal{D}_U = \{\tilde{\mathbf{x}}\}_{i=1}^{|\mathcal{D}_U|}$.

An overview of the four learning paradigms evaluated in this study—supervised learning, self-training, CLIP-style contrastive learning, and NegCLIP—is provided in Figure 2. For neural network used for training, we denote the ECG encoder as $f_\theta :\rightarrow \mathbb{R}^d$ with linear classifier for supervised and self-training and MLP projection layer for CLIP based training (He et al., 2016). We also define text encoder as $h_\phi :\rightarrow \mathbb{R}^d$ for CLIP based training. $\theta$ and $\phi$ are learnable parameters. In practice, the ECG encoder is a 1D ResNet-50 backbone followed by either a linear classification head for supervised and self-training or MLP projection head for contrastive learning. For CLIP-style training, both ECG and text embeddings are normalized. We now describe each training paradigm in detail.

**Supervised training.** The baseline model is trained end-to-end on expert over-read labels $\mathbf{y}_i$ using a binary cross-entropy (BCE) loss with class-balanced weight:

$$
\mathcal{L}_{sup} = -\frac{1}{|\mathcal{D}_L| \times |C|} \sum_{i=1}^{\mathcal{D}_L} \sum_{c=1}^{C} [w^c \cdot \mathbf{y}_i^c \log \sigma(f_\theta(\mathbf{x}_i^{\text{weak}})) + (1 - \mathbf{y}_i^c) \log(1 - \sigma(f_\theta(\mathbf{x}_i^{\text{weak}})))],
$$
$$
w^c = \min \left( \frac{\sum_{i=1}^{N}(1 - \mathbf{y}_i^c)}{\sum_{i=1}^{N} \mathbf{y}_i^c}, \, w_{\max} \right)
$$
(1)

where $\sigma$ is the sigmoid function, $w^c$ is the positive class weight for class $c$, and $w_{\max}$ is a maximum cap to prevent extreme weights for highly imbalanced classes. We cap the positive class weight at a fixed maximum value to avoid unstable gradients for extremely rare diagnoses while still mitigating severe class imbalance. Additionally, we applied weak augmentation on ECG data for fair comparison with Self-training method describe after

**Self-training.** To incorporate public ECGs without expert over-read labels, we employ a FixMatch-style framework (Sohn et al., 2020). The same architecture used in the supervised setting is trained jointly on labeled and unlabeled data. For labeled ECGs, we apply a weak augmentation and compute the supervised loss in the same manner as our supervised baseline. For unlabeled ECGs, the model first processes a weakly augmented view to obtain pseudo-labels $\mathbf{p}_i = \sigma(f_\theta(\tilde{\mathbf{x}}_i^{\text{weak}}))$. We apply a single confidence threshold $\alpha$ shared across all diagnostic labels, masking out unlabeled samples whose maximum predicted probability does not exceed this threshold. The retained pseudo-labels are then used as regression targets for the corresponding strongly augmented views of the same unlabeled ECGs, following the FixMatch formulation based on masked binary cross-entropy. The overall self-training loss is defined as:

$$\mathcal{L}_{self} = \mathcal{L}_{sup} + \lambda \mathcal{L}_{unsup},$$

$$\mathcal{L}_{unsup} = -\frac{1}{|\mathcal{D}_U| \times |C|} \sum_{i=1}^{\mathcal{D}_U} \sum_{c=1}^{C} \mathbf{1}(\sigma(f_\theta(\tilde{\mathbf{x}}_i)) \geq \alpha) \left[ p_i \log \sigma(f_\theta(\tilde{\mathbf{x}}_i^{\mathrm{str}})) + (1-p_i) \log(1 - \sigma(f_\theta(\tilde{\mathbf{x}}_i^{\mathrm{str}}))) \right],$$

(2)

where $\lambda$ balances the supervised and unsupervised loss terms and $\tilde{\mathbf{x}}_i^{\mathrm{str}}$ is strongly augmented $\tilde{\mathbf{x}}_i$. Unsupervised term here supposed to work like regularization term during the training. (Sohn et al., 2020).

**CLIP-based training.** To explore contrastive learning based approach on paired ECG and text data, we train a CLIP-style model that aligns ECG signals and expert over-read text (Radford et al., 2021). We split over-read reports into individual diagnosis statements, treating each statement as a positive pair for the corresponding ECG for fine-grained supervision. This produces multiple diagnosis statements per ECG, allowing each ECG to form multiple positive ECG-text pairs during training. This diagnosis level formation encourages fine-grained alignment between ECG signals and specific clinical concepts. The CLIP loss is defined as:

$$\mathcal{L}_{\mathrm{CLIP}} = -\frac{1}{2|\mathcal{D}_L|} \left[ \sum_{i=1}^{\mathcal{D}_L} \log \frac{\exp(s_{ii})}{\sum_{j=1}^{\mathcal{D}_L} \exp(s_{ij})} + \sum_{j=1}^{\mathcal{D}_L} \log \frac{\exp(s_{jj})}{\sum_{i=1}^{\mathcal{D}_L} \exp(s_{ji})} \right]$$

(3)

where $s_{ij} = f_\theta(\mathbf{x}_i)^\top h_\phi(\mathbf{t}_j)/\tau$ is the cosine similarity between ECG $\mathbf{x}_i$ and text embeddings $\mathbf{t}_j$ scaled by a learnable temperature $\tau$.,

**NegCLIP-based training.** To fully exploit the paired nature of our dataset, we extend CLIP with diagnosis-level hard negatives derived from machine-read vs. over-read discrepancies (Yuksekgonul et al., 2023). We treat all over-read statements as ground-truth positives and any statement that appears in the machine-read report but not in the over-read as an explicit hard negative. The NegCLIP loss is defined as:

$$\mathcal{L}_{\mathrm{neg\text{-}CLIP}} = -\frac{1}{2|\mathcal{D}_L|} \left[ \sum_{i=1}^{\mathcal{D}_L} \log \frac{\exp\left(s_{ii}\right)}{\sum_{j=1}^{\mathcal{D}_L} \exp\left(s_{ij}\right) + \exp\left(\hat{s}_{ij}\right)} + \sum_{j=1}^{\mathcal{D}_L} \log \frac{\exp\left(s_{jj}\right)}{\sum_{j=1}^{\mathcal{D}_L} \exp\left(s_{ji}\right) + \exp\left(\hat{s}_{ji}\right)} \right]$$

(4)

where $\hat{s}_{ij} = f_\theta(\mathbf{x}_i)^\top h_\phi(\hat{\mathbf{t}}_j)/\tau$ is the similarity between ECG $\mathbf{x}_i$ and machine-read text $\hat{\mathbf{t}}_j$. This design pushes ECG embeddings away from machine-only textual hypotheses while pulling them toward expert-confirmed diagnoses, thereby leveraging the unique expert/machine pairing in our corpus.

## 4. Experiments

### 4.1. Dataset curation and label extraction

**Data and Preprocessing** We used a combination of private and public electrocardiogram (ECG) datasets to investigate the role of expert over-read data in automated ECG

interpretation. Our primary corpus is a private dataset collected at Weill Cornell Medicine / NewYork–Presbyterian Hospital, containing approximately two million standard 12-lead ECGs, each with both the machine interpretation and a expert over-read provided by experts with more than 30 years of experience. All recordings were acquired at 250 Hz for 10 s and serve as the foundation for supervised and contrastive training. An overview of the full dataset flow is provided in Figure A.1.

To complement these data with additional diversity and unlabeled samples, we incorporated two public datasets: (i) the **PhysioNet 2021 Challenge** dataset (Reyna et al., 2021), using six constituent sources (CPSC, CPSC-Extra, PTB-XL, Georgia, Ningbo, Chapman) and excluding PTB and St. Petersburg INCART due to short duration or inconsistent sampling rates; and (ii) the **MIMIC-IV-ECG v1.0** database (Gow et al., 2023), consisting of 10-s recordings sampled at 500 Hz. These public datasets were used primarily for self-training method.

All signals passed through a standardized preprocessing pipeline to ensure temporal and spectral consistency. ECGs with invalid values (e.g., `NaN`, constant-zero leads) were removed. Signals below 250 Hz were discarded, and those above 250 Hz were downsampled via linear interpolation. Longer recordings were split into non-overlapping 10-s windows. After temporal alignment, we applied: (1) baseline-wander removal with a 0.5 Hz high-pass filter, (2) per-lead $z$-score normalization, and (3) a 50–60 Hz notch filter to suppress power-line interference. All final waveforms were represented as 12-lead, 10-s signals at 250 Hz for downstream analyses.

**Label extraction.**  Each ECG record in our private dataset contains two independent textual interpretations: the initial machine-generated report and the expert over-read diagnosis. To enable quantitative evaluation, we converted these free-text descriptions into a structured multi-label representation covering the most clinically relevant rhythm and morphological abnormalities. A detailed summary of label frequencies for both machine-read and over-read sources is provided in Table A.1.

In consultation with cardiologists, we defined a vocabulary of 13 diagnostic labels spanning rhythm abnormalities, conduction blocks, axis deviations, and ischemic findings. These categories reflect conditions routinely reported in clinical cardiology practice and capture the most prevalent and clinically actionable ECG patterns in our dataset.

To ensure consistent label assignment, we curated a synonym map that consolidated equivalent medical expressions, abbreviations, and spelling variants into a unified terminology. For example, phrases such as "normal sinus rhythm" and "NSR" were grouped under *Sinus Rhythm*, while terms like "left anterior fascicular block" and "left axis deviation" were standardized to *Left Axis Deviation*. This mapping was iteratively refined with cardiologist input to guarantee alignment between machine-read and expert over-read interpretations.

The final text-processing pipeline was applied identically to both machine-generated and expert reports, producing a harmonized label vocabulary across sources. Each ECG was ultimately represented as a binary vector of 13 diagnostic labels, enabling direct, systematic comparison of model performance on machine-read versus expert-validated interpretations.

### 4.2. Implementation Details and Evaluation

All experiments use a unified 1D ResNet–50 ECG encoder (He et al., 2016) and the AdamW optimizer with a learning rate of $1 \times 10^{-4}$, weight decay of $1 \times 10^{-4}$, and cosine annealing over 100 epochs. Supervised and self-training models use a batch size of 512, while contrastive models use a batch size of 256 paired ECG–text samples. Early stopping is applied based on validation macro-AUPRC with a patience of 10 epochs. The dataset is divided into 80% development and 20% testing, with the development portion further split into 80% training and 20% validation. A fixed random seed ensures identical splits across all training paradigms.

**Supervised.** Supervised training uses BCEWithLogitsLoss with class-balanced positive weights (ratio of negatives to positives, capped at 50) and applied same weak augmentaiton that used for the self-training method. Model selection is based on validation macro-AUPRC.

**Self-training.** For self-training, labeled samples use the same preprocessing as supervised training. For unlabeled ECGs, weak augmentations include Gaussian noise ($\sigma = 0.015$), amplitude scaling, and small temporal shifts. Strong augmentations include time warping, temporal masking, random resampling, lead dropout, and Gaussian noise with $\sigma = 0.02$ (Raghu et al., 2022; Sohn et al., 2020). A single confidence threshold ($\alpha = 0.95$) is used to filter pseudo-labels, ensuring that only highly reliable predictions contribute to the unsupervised loss. We choose ($\lambda = 1$) by tuning it on validation performance of AUPRC metric which demonstrate at Figure A.3.

**CLIP and NegCLIP.** For contrastive learning, the ECG encoder is paired with the original CLIP text transformer (Radford et al., 2021). ECG and text representations are projected into a shared 512-dimensional space and $\ell_2$-normalized. Over-read reports are decomposed into diagnosis-level statements, enabling multi-positive alignment between each ECG and the set of diagnoses appearing in the corresponding expert report. NegCLIP extends this setup by incorporating machine-read–only statements as explicit hard negatives (Yuksekgonul et al., 2023). Both contrastive variants use identical optimization settings for fair comparison.

## 5. Results

### 5.1. Evaluation Results

We evaluate all models using per-label AUROC and AUPRC, capturing both overall discrimination and precision under substantial class imbalance. We report AUROC/AUPRC as mean [95% CI], with bootstrapping confidence intervals on the test set. For supervised and self-training models, these metrics are computed from sigmoid prediction scores, whereas for CLIP and NegCLIP, the ECG–text cosine similarity values (ranging from $-1$ to 1) serve directly as prediction scores for each diagnostic label. Table 1 provides an aggregated performance comparison across the four learning paradigms—supervised learning, self-training, CLIP, and NegCLIP—and serves as the primary reference for the cross-paradigm performance trends discussed below. For completeness, macro-averaged ROC and precision–recall curves for all paradigms are included in Figure A.2.

Table 1: **Overall comparison across all paradigms.** Per-label performance (AUROC and AUPRC), reported in percentage (%), for supervised, self-training, CLIP, and NegCLIP models evaluated against expert over-read labels. Results are shown as mean [95% CI]. Best performance per metric per row is shown in **bold**, and second best is underlined.

| Diagnosis | Supervised | | Self-Training | | CLIP | | NegCLIP | |
|---|---|---|---|---|---|---|---|---|
| | AUROC | AUPRC | AUROC | AUPRC | AUROC | AUPRC | AUROC | AUPRC |
| Sinus Rhythm | 99.7[99.7–99.7] | 99.9[99.9–99.9] | **99.7**[99.7–99.7] | **99.9**[99.9–99.9] | 89.5[89.3–89.6] | 98.6[98.6–98.6] | **98.9**[98.8–99.0] | 99.8[99.8–99.8] |
| RBBB | **99.9**[99.9–99.9] | 99.0[98.9–99.1] | 99.9[99.9–99.9] | **99.0**[99.0–99.1] | 99.7[99.7–99.7] | 95.2[94.9–95.4] | 99.9[99.9–99.9] | 98.1[97.9–98.3] |
| LBBB | **99.9**[99.9–99.9] | 98.2[98.1–98.4] | 99.9[99.9–99.9] | **98.3**[98.1–98.4] | 99.5[99.5–99.5] | 79.4[78.4–80.2] | 99.9[99.9–99.9] | 97.5[97.2–97.7] |
| AFib | 99.8[99.8–99.8] | **97.1**[97.0–97.3] | **99.8**[99.8–99.8] | 97.3[97.2–97.5] | 87.6[87.4–87.8] | 56.7[56.1–57.3] | 99.8[99.8–99.8] | 96.3[96.1–96.6] |
| Atrial Flutter | 98.8[98.7–98.9] | 82.7[81.9–83.5] | **98.9**[98.8–99.0] | **84.8**[84.1–85.5] | 62.5[61.9–63.0] | 8.30[7.70–8.90] | 98.9[98.8–99.0] | 75.9[74.9–76.9] |
| Left AD | 99.4[99.3–99.4] | 94.6[94.5–94.8] | **99.4**[99.4–99.4] | **94.8**[94.6–95.0] | 97.6[97.6–97.7] | 73.3[72.8–73.8] | 98.8[98.8–98.8] | 88.6[88.2–88.9] |
| Right AD | 99.7[99.6–99.7] | 88.2[87.5–88.9] | **99.7**[99.6–99.7] | **89.5**[88.9–90.1] | 99.4[99.4–99.5] | 80.9[80.1–81.9] | 99.6[99.5–99.6] | 86.3[85.7–87.1] |
| Right SAD | 99.8[99.8–99.9] | 74.9[72.9–77.0] | **99.8**[99.8–99.9] | **77.8**[76.0–79.6] | 99.2[99.0–99.3] | 46.3[44.0–48.7] | 99.8[99.7–99.8] | 70.6[68.3–72.8] |
| LVH | 94.4[94.3–94.5] | 78.7[78.4–79.0] | **94.6**[94.5–94.7] | **79.5**[79.2–79.8] | 90.5[90.4–90.6] | 66.7[66.3–67.0] | 93.4[93.3–93.5] | 73.3[72.9–73.7] |
| RVH | 99.3[99.2–99.3] | 79.1[78.3–79.9] | **99.3**[99.2–99.4] | **81.6**[80.8–82.3] | 98.5[98.4–98.6] | 66.1[65.1–67.1] | 99.1[99.0–99.2] | 77.8[76.9–78.6] |
| Anterior MI | **97.7**[97.7–97.7] | 88.0[87.7–88.2] | 97.7[97.6–97.7] | **88.2**[88.0–88.5] | 96.0[95.9–96.1] | 81.8[81.5–82.1] | 96.9[96.8–97.0] | 81.6[81.2–81.9] |
| Inferior MI | **98.8**[98.8–98.9] | 90.9[90.7–91.2] | 98.8[98.8–98.8] | **91.0**[90.7–91.2] | 98.4[98.4–98.5] | 88.3[88.0–88.6] | 98.4[98.4–98.5] | 87.8[87.4–88.1] |
| Posterior MI | 98.6[98.5–98.8] | 63.9[62.6–65.2] | **98.7**[98.5–98.8] | **65.0**[63.7–66.2] | 97.8[97.6–97.9] | 51.8[50.4–53.2] | 98.5[98.4–98.7] | 60.2[58.9–61.6] |

**Supervised Learning Baseline** The supervised results in Table 1 show that models trained on expert over-read labels and evaluated against the same ground truth achieve the strongest overall performance. These models form the baseline against which all other learning paradigms are compared. To contextualize the advantages of over-read supervision, we provide an ablation study in Section 5.2 and Table 2 that compares the same model architecture trained instead on machine-read labels. This experiment mimics common practice in prior ECG work and demonstrates that models trained on machine-read labels exhibit inflated performance when evaluated against machine-read test labels, but suffer notable drops when evaluated against expert over-read ground truth. As shown in Table 2, evaluating a machine-trained model on machine-read test labels can substantially overstate performance, since the model is effectively judged against the same noisy labeling process it learned. In contrast, the more clinically meaningful estimate is the same machine-trained model evaluated on expert over-read labels, which better reflects its true expert-aligned performance.

**Self-Training with Unlabeled Public Data** Self-training results in Table 1 demonstrate that incorporating unlabeled ECGs from public datasets through a FixMatch-style framework improves performance across nearly all diagnoses. By expanding the training signal beyond the expert-labeled private dataset, the model benefits from better representation of diverse ECG patterns and improves generalization, particularly for rare or underrepresented classes. The gains are most prominent in AUPRC, highlighting the benefit of leveraging large-scale unlabeled corpora under extreme class imbalance.

**Contrastive Learning with CLIP and NegCLIP** Table 1 also reports the contrastive learning results. Standard CLIP, which aligns ECG signals purely with expert over-read text, achieves competitive discrimination but struggles with precision for rare diagnoses. NegCLIP, which additionally leverages discrepancies between machine-read and over-read text as hard negatives, consistently improves both AUROC and AUPRC. Incorporating these disagreement-driven negatives sharpens the model's ability to differentiate subtle diagnostic

Table 2: **Supervised learning.** Per-diagnosis AUROC/AUPRC for identical models trained on machine-read vs. expert over-read labels. Machine-trained models are evaluated against both machine-read and over-read test labels, while over-read–trained models are evaluated against over-read labels, highlighting that machine-only evaluation can overstate expert-aligned performance.

| Diagnosis | Trained on Machine-read | | | | Trained on Over-read | |
|---|---|---|---|---|---|---|
| | AUROC–Machine | AUPRC–Machine | AUROC–Over | AUPRC–Over | AUROC | AUPRC |
| Sinus Rhythm | 0.989 | 0.998 | 0.996 | 0.999 | **0.997** | **0.999** |
| RBBB | **0.999** | 0.987 | 0.999 | 0.989 | 0.999 | **0.990** |
| LBBB | 0.999 | 0.971 | 0.998 | 0.969 | **0.999** | **0.982** |
| Atrial Fibrillation | 0.994 | 0.919 | 0.997 | 0.949 | **0.998** | **0.971** |
| Atrial Flutter | 0.980 | 0.696 | 0.966 | 0.691 | **0.988** | **0.827** |
| Left Axis Deviation | 0.993 | 0.944 | 0.993 | 0.941 | **0.994** | **0.946** |
| Right Axis Deviation | **0.997** | 0.868 | 0.989 | 0.774 | 0.997 | **0.882** |
| Right Superior AD | **0.999** | **0.759** | 0.988 | 0.521 | 0.998 | 0.749 |
| LVH | **0.949** | 0.779 | 0.930 | 0.756 | 0.944 | **0.787** |
| RVH | **0.995** | 0.775 | 0.849 | 0.507 | 0.993 | **0.791** |
| Anterior MI | 0.976 | **0.889** | 0.962 | 0.781 | **0.977** | 0.880 |
| Inferior MI | 0.988 | 0.906 | 0.984 | 0.884 | **0.988** | **0.909** |
| Posterior MI | **0.992** | 0.550 | 0.943 | 0.328 | 0.986 | **0.639** |

cues, bringing performance closer to the supervised baseline while retaining the flexibility of a multimodal formulation. An additional experiment evaluating the impact of text granularity—independent diagnosis statements versus whole report encoding—is presented in Section 5.2.

## 5.2. Ablation Studies

**Supervised: Machine-read vs. Over-read Labels.** To further investigate how label source affects supervised training, we evaluate the same model architecture trained on machine-read versus expert over-read labels shown in Table 2. Models trained on machine-read labels appear highly accurate when evaluated against machine-read test labels, but their precision drops when evaluated against expert ground truth, revealing the noise and bias in automated interpretations. Because public machine-read datasets differ in sampling rate and signal format from our 250 Hz private ECGs, we replicated this setting by training on our own machine-read labels. Overall, this ablation shows that machine-read labels can give a misleading impression of performance, whereas expert over-reads provide far more reliable supervision.

**Independent Diagnosis Text vs. Whole-text Formulation.** In addition to the main NegCLIP formulation, we conducted an ablation to evaluate whether diagnosis-level text granularity is necessary for effective multimodal contrastive learning. Instead of splitting expert and machine-read reports into individual diagnosis statements, a whole-text variant encodes each report as one full sequence using the same BERT encoder (Devlin et al., 2019). As shown in Figure 3, this whole-text approach performs substantially worse across nearly all diagnoses. Because most diagnoses in an ECG report are unrelated to each other and machine–expert discrepancies typically occur in only one or two labels, collapsing the

entire report into a single embedding dilutes these fine-grained disagreement signals. In contrast, our diagnosis-independent formulation preserves diagnosis-specific variation and yields markedly stronger supervision.

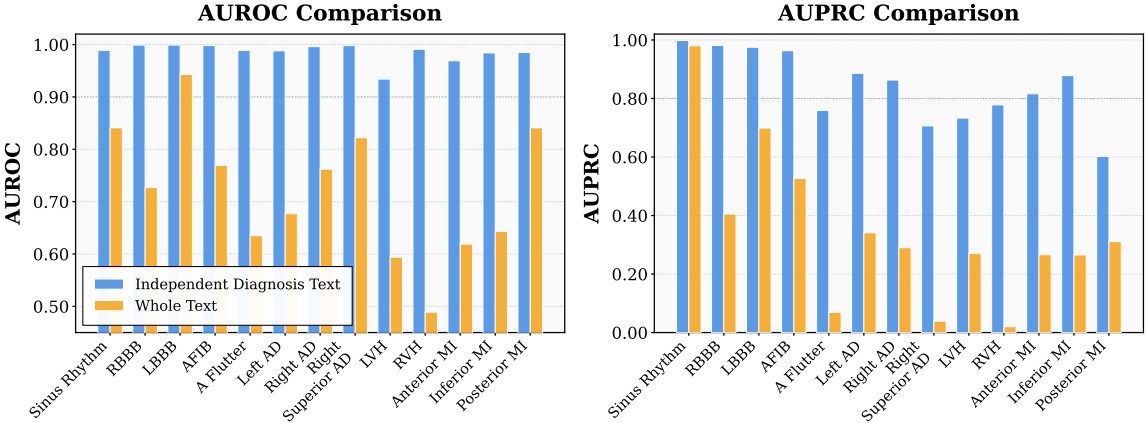

Figure 3: **Impact of Diagnosis-level Text Granularity.** Comparison of per-label ARUOC AUPRC when using independent diagnosis statements versus whole-text report embeddings in multimodal contrastive learning. Independent diagnosis text provides substantially stronger supervision across most diagnostic categories.

## 6. Discussion

Our study underscores the critical importance of expert over-read ECG data for developing clinically reliable AI systems and was designed to explicitly examine the risks of relying on machine generated ECG interpretations as primary supervision. We demonstrate that models trained on cardiologist-verified over-reads achieve consistently higher fidelity than those trained on machine-read interpretations, particularly for rare but clinically significant diagnoses where precision is most critical. By leveraging a uniquely large paired dataset of over-read and machine-read ECGs, we systematically evaluate four methods—supervised, self-training, CLIP, and NegCLIP—and show that incorporating over-read supervision not only boosts diagnostic accuracy but also exposes systematic biases embedded in machine-read interpretations. These results highlight that strong performance achieved using machine-read supervision can be misleading when evaluated against similar machine generated labels. Beyond supervised learning, our results demonstrate that expert knowledge can be propagated in scalable ways. Self-training extends expert supervision to large unlabeled public ECG datasets, improving generalization under extreme class imbalance, while NegCLIP introduces a multimodal pathway that explicitly exploits discrepancies between machine and expert interpretations to learn more clinically aligned representations. Importantly, these approaches are intended to amplify expert supervision rather than replace expert over-reads, enabling models to benefit from expert judgment in data-limited settings.

This work also has important limitations that delineate where our methods may perform poorly. Because our data originate from a single health system, model performance may

degrade under domain shifts in acquisition protocols, patient populations, or reporting styles. Our label extraction relies on structured mappings from free-text reports, which may miss nuanced clinical qualifiers and limit performance for subtle diagnostic distinctions. Finally, our study implicitly assumes that expert over-reads provide a consistently more reliable supervisory signal than vendor machine-read interpretations. In practice, device algorithms may improve over time, narrowing the gap and reducing the relative benefit of discrepancy-aware objectives. Moreover, expert over-reads are not error-free. Clinician corrections can still be affected by inter-reader variability, fatigue, or ambiguous ECG patterns, which may introduce residual noise into the ground-truth labels.

## 7. Conclusion

We show that expert cardiologist over-reads provide a stronger and more clinically reliable supervisory signal than machine-generated ECG interpretations. Using a large paired corpus of over two million ECGs with both machine-read and expert over-read reports, we demonstrate that models trained with expert supervision consistently outperform those trained on machine labels across supervised, self-training, and multimodal contrastive learning paradigms, with the largest gains observed for rare and clinically important diagnoses. Our results further reveal that reliance on machine-read labels can produce misleadingly optimistic evaluations when tested against similarly noisy ground truth. By leveraging unlabeled public ECGs and explicitly modeling machine–expert discrepancies, we present scalable strategies to propagate expert knowledge beyond limited annotated data. Together, these findings establish expert over-reads as a critical foundation for developing trustworthy and clinically aligned ECG AI systems.

## Acknowledgments

This work was supported by funding from NewYork-Presbyterian for the NYP-Cornell Cardiovascular AI Collaboration.

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

## Appendix A. Data Flow

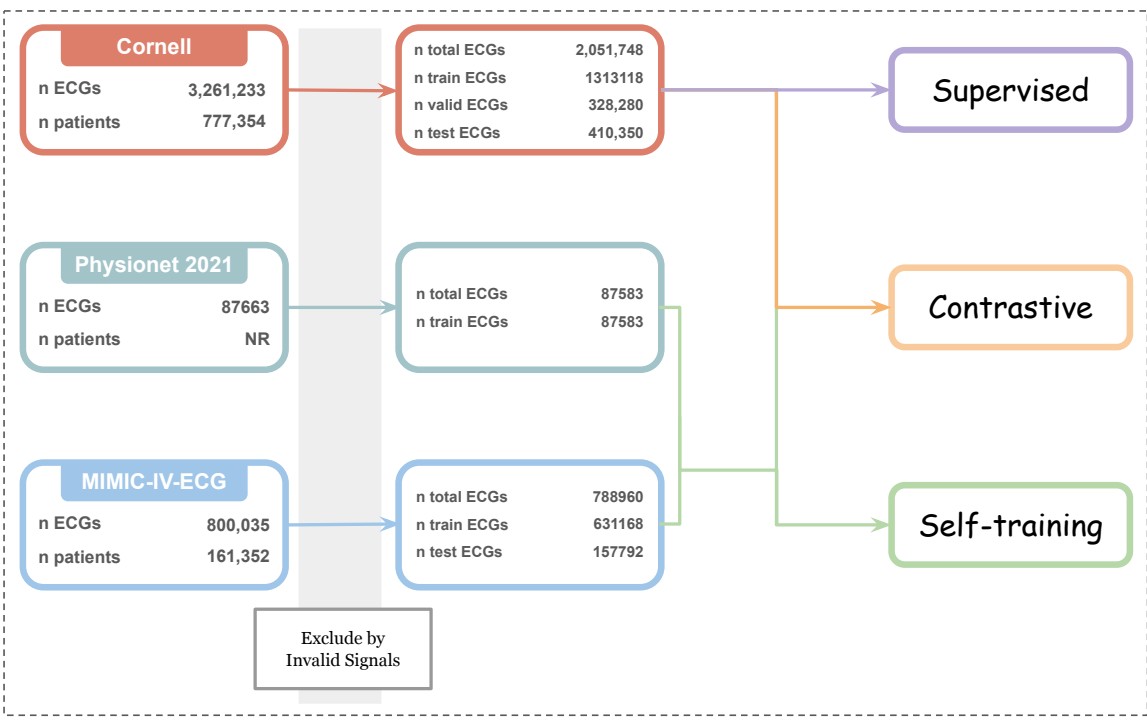

Figure A.1: **Data Flow of Private and Public Datasets.** Overview of dataset composition and flow used in our experiments, including the Cornell private ECG dataset, MIMIC-IV-ECG, and the PhysioNet 2021 Challenge dataset.

## Appendix B. Cornell Private Data

Table A.1: **Cornell Private Dataset Summary.** Comparison of machine-read and expert over-read label counts. The Disagreement column reports the total number of studies where the two sources differ. *+Added%* column indicates the proportion of additional positive diagnoses introduced by experts—cases that were absent in the machine-read output but marked positive in the expert over-read. *-Removed%* column reflects positive labels removed by experts—cases that were labeled positive by the machine-read system but negated in the over-read. Together, these metrics show how expert review modifies automated ECG interpretations in both directions.

| Diagnosis | Machine-read | Over-read | Disagreement | +Added % | -Removed % |
|---|---|---|---|---|---|
| Sinus Rhythm | 1,826,249 | 1,848,631 | 43,848 | +1.81% | -0.59% |
| RBBB | 130,380 | 137,237 | 10,267 | +6.57% | -1.31% |
| LBBB | 52,687 | 59,449 | 11,678 | +17.50% | -4.67% |
| Atrial Fibrillation | 148,971 | 144,652 | 25,855 | +7.23% | -10.13% |
| Atrial Flutter | 25,566 | 43,071 | 26,189 | +85.45% | -16.98% |
| Left Axis Deviation | 198,282 | 205,834 | 12,554 | +5.07% | -1.26% |
| Right Axis Deviation | 37,367 | 50,708 | 21,383 | +46.46% | -10.76% |
| Right Superior AD | 5,090 | 9,019 | 5,491 | +92.53% | -15.34% |
| LVH | 270,853 | 315,212 | 64,415 | +20.08% | -3.70% |
| RVH | 13,157 | 42,192 | 31,579 | +230.35% | -9.67% |
| Anterior MI | 290,263 | 246,358 | 114,729 | +12.20% | -27.33% |
| Inferior MI | 179,658 | 179,836 | 54,542 | +15.23% | -15.13% |
| Posterior MI | 7,183 | 28,908 | 26,035 | +332.45% | -30.00% |

## Appendix C. Overall Comparison AUROC and AUPRC Curves

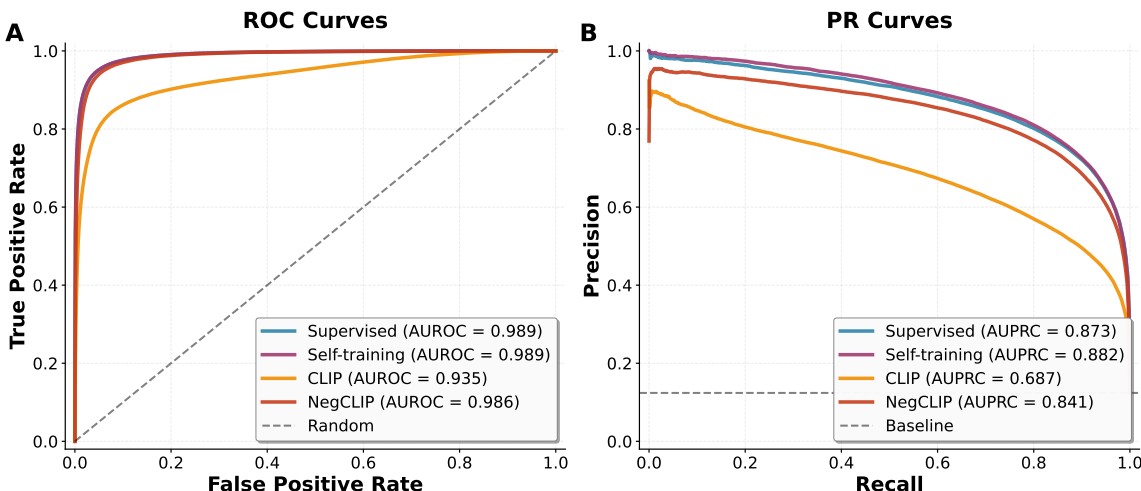

Figure A.2: **Overall comparison of AUROC and AUPRC curves across paradigms.** Panel (A) shows macro-averaged ROC curves (AUROC) for supervised, self-training, CLIP, and NegCLIP models. Panel (B) shows macro-averaged precision–recall curves (AUPRC) for the same paradigms.

## Appendix D. Effect of $\lambda$ on Self-Training Performance

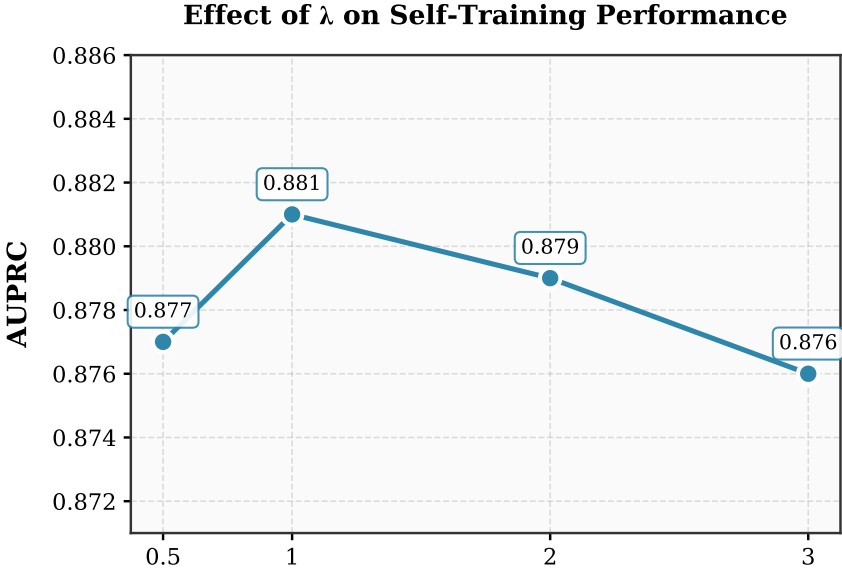

Figure A.3: **Tuning $\lambda$ on Self-Training Performance.** macro-AUPRC performance change over different lambda values. We picked best performing parameter for our evaluation.

### Appendix E. Dataset Scaling Experiment

To quantify how unlabeled public ECGs contribute to self-training, we conduct a dataset scaling experiment in which increasing fractions of the public corpus are incorporated into the self-training pipeline. For each subset size, we evaluate AUPRC on the held-out test set. As shown in Figure A.4, model performance improves consistently as more unlabeled data is included, reflecting improved representation of low-prevalence ECG phenotypes. These results demonstrate that large-scale unlabeled corpora play a critical role in amplifying the benefits of expert over-read supervision when combined with pseudo-label–based training.

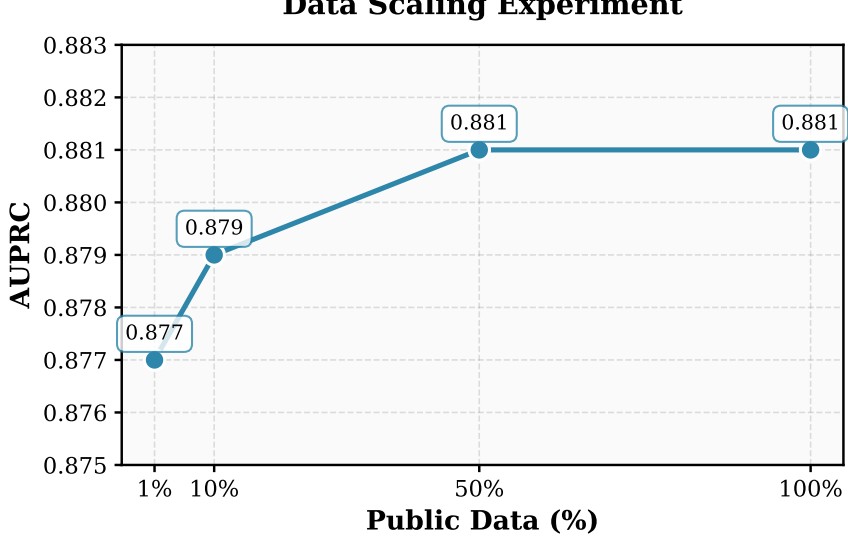

Figure A.4: **Public Dataset Scale on Self-Training Performance.** AUPRC improves steadily as larger portions of public ECG data are incorporated into the self-training pipeline.

## Appendix F. Label Acronyms

Table A.2: **Explanation of Label Acronyms.** Full terminology corresponding to diagnostic abbreviations used throughout this paper.

| Acronym / Term | Full Description |
| --- | --- |
| RBBB | Right Bundle Branch Block |
| LBBB | Left Bundle Branch Block |
| AD | Axis Deviation |
| LVH | Left Ventricular Hypertrophy |
| RVH | Right Ventricular Hypertrophy |
| MI | Myocardial Infarction |
| Anterior MI | Anterior Myocardial Infarction |
| Inferior MI | Inferior Myocardial Infarction |
| Posterior MI | Posterior Myocardial Infarction |
| AFib | Atrial Fibrillation |

