# OpenReview forum: "Beyond Machine Interpretation: Learning from Expert Over-Reads Improves ECG Diagnosis"
_MIDL.io/2026/Conference — MIDL 2026 Poster_

### Official Review · Reviewer_btF2 · 2026-01-06

**Confidence:** 5
**Preliminary Rating:** 2
**Final Rating:** 2

**Summary:**

This paper benchmarks many different representation learning methods that could be used for ECG classification tasks. They prepare a large corpus of ECG signals. They also try to show that "expert-supervised models consistently outperform those
trained on machine-read labels"
..........................

**Strengths:**

The paper discusses interesting topics working with difficult data.
The work tackles many downstream tasks.
...............................................................................................

**Weaknesses:**

The results presented have no standard deviation reported so it is not clear how to assess significance to support the claims made about self-training performing better than supervised models.

Please reduce the reliance on the appendix figures and text. The paper should be self contained.
The 3rd contributions is studying "Machine-read vs. Over-read Labels" but the supporting figures for this are in the appendix. Everything needed to directly support the contributions should be in the main text.

The figure captions should help me to read and understand the figures. For most plots I don't see clear guidance on what I should be looking at. I'm also not sure why curves are needed when a table would achieve the same thing.

**Detailed Comments:**

It is unclear why we should care about the difference between models trained on machine reads and a model trained on expert reads. The intro states that machine reads do not perform better than humans, so then it makes sense they would not be as good for training. So the need here is just for better performing models.

The paper might be better positioned as just trying to improve performance on ECG using semi-supervised models.

**Justification Of Final Rating:**

I still think the framing should be more straightforward that this comparison is between models trained on automated labels vs human created labels.
Also this conference is for medical imaging and this work focuses on ECG signals.

**Justification Of The Preliminary Rating:**

Paper not self contained.
Contributions unclear.
Motivation unclear.
......................................................................................................................................

**Questions To Address In The Rebuttal:**

Is it public? If so please add a URL where it can be found if it will be considered a contribution.
If not then I'm not sure this should be considered a contribution, what is it a contribution to?

---

> ### Author Response · Authors · 2026-01-24
>
> Dear Reviewer btF2,
>
> Thank you for your thorough review for our work. We carefully read all of your comments and made our best effort to address each of them. Some points have been merged where they concern closely related subjects.
>
> ---
>
> ### Results without uncertainty
>
> Thank you for raising this point. We agree that reporting uncertainty is important. Accordingly, we added 95% confidence intervals to the results reported in Table 1, which summarizes the overall performance across all paradigms.
>
> ---
>
> ### Reliance on the Appendix
>
> We agree with this concern. In this revision, we were able to extend our paper to 12 pages, which allowed us to include Machine-read vs. Over-read comparison tables to the main text as Table 2. Additionally, we removed a couple appendices that could be overlapped with figures in main-text to reduce the reliance on the Appendix. We reorganized the figures to make the main text self-contained, with appendices now serving a supplementary role.
>
> ---
>
> ### Better figure configuration/caption
>
> We understand your concern regarding figure configuration and captions. We revised captions to improve clarity and readability. In particular, we replaced Figure 1 with a new figure and caption to demonstrate the clear workflow of acquiring machine-read and over-read labels. Also, we moved our Table 2 to the main text. Lastly we also replaced the Figure 3 from ROC/PR Curve to simple bar plot to clearly demonstrate the difference between text granularity performance.
>
> ---
>
> ### Why should care about model trained machine-read and over-read
>
> This work is strongly motivated by clinical practice. Not only many non-cardiology clinicians highly rely on machine-reading for ECG, but also most of the open sourced models and public data are solely relying on machine-read interpretation. We are trying to demonstrate that ECG interpretation relying on machine-read can underperform in real clinic scenarios, while a model trained on over-read shows better performance against the over-read labels which is close to what expert cardiologists see from ECG reading.
>
> ---
>
> ### Public code and models
>
> Thank you for pointing this out. We have released our code and trained model weights on github
> https://github.com/tyoung089/Overread_ecg

---

### Official Review · Reviewer_Wcvu · 2026-01-14

**Confidence:** 3
**Preliminary Rating:** 5
**Final Rating:** 5

**Summary:**

The work contributes by curating a large-scale ECG dataset with both machine-read and expert ("over-read") labels. It also compares different self-supervised (SSL) and semisupervised learning strategies (contrastive learning, CLIP/negCLIP and FixMatch) and discusses their advantages and limitations.

**Strengths:**

- All experiments and the datasets are very well designed and thoroughly executed
- The analyses are interesting insights, especially in the high value of NegCLIP
- The results promise clinical value of leveraging scalable learning strategies for differently labelled (sub)-datasets

**Weaknesses:**

- there are only minor points that could be criticised, mainly whether the paper would not have fit better in the validation paper track?
- the supervised ResNet (on over-reads) was trained w/o augmentation - is this the best option?

**Detailed Comments:**

Since the authors specifically mention benefits in data-limited scenarios an ablation of available dataset sizes (subsampling) could add some extra value

**Justification Of Final Rating:**

The rebuttal has addressed some remaining points of criticism and the paper is now ready to be accepted, therefore I keep my previous rating and justification "While there is no inherent strong technical novelty, the paper provides meaningful insights and its experiments are very thoroughly performed using large-scale datasets."

**Justification Of The Preliminary Rating:**

While there is no inherent strong technical novelty, the paper provides meaningful insights and its experiments are very thoroughly performed using large-scale datasets. (note this was an emergency review, hence the brevity)

**Questions To Address In The Rebuttal:**

Whether code and trained models will be made available?

---

> ### Comment · Reviewer_btF2 · 2026-01-17
>
> I don't see in your review where you comment on the main stated contribution of the paper: "models trained
> with expert over-read data consistently outperform those trained on machine-read labels" which digging into the paper I understand is: comparing a model trained on automatically predicted labels (by inferior models) vs human predicted labels.
>
> I think this contribution of the paper is misleading as it sounds like something different than it is. "expert over-read labels" sounds like multiple experts reviewed the labels vs a single expert, but the experiments are not that.

---

> > ### Author Response · Authors · 2026-01-19
> >
> > Dear Reviewers ,
> >
> > First, we thank Reviewer btF2 for the comment. We would like to provide a brief clarification regarding the term over-read. Over-reading (or over-read) is a commonly used term in clinical practice and the medical literature, referring to cardiologists reviewing machine-generated ECG interpretations to ensure accurate interpretation and diagnosis (Smulyan, 2019; Schläpfer and Wellens, 2017). In our work, the term over-read does not refer to multi-expert or consensus readings. We will further clarify this point in the official rebuttal revision.
> >
> > We stated that “to address this challenge, we curated a large corpus of over two million 12-lead ECGs, each containing both the original machine-read interpretation and a cardiologist over-read provided by experts with more than 30 years of experience.” The over-read labels in our dataset were obtained through a standardized process in which experienced cardiologists systematically reviewed and corrected machine-read interpretations.
> >
> > We will clarify this description more explicitly in the upcoming revised rebuttal
> >
> > Thank you.
> >
> > Smulyan, Harold. "The computerized ECG: friend and foe." The American journal of medicine 132.2 (2019): 153-160.
> > Schläpfer, Jürg, and Hein J. Wellens. "Computer-interpreted electrocardiograms: benefits and limitations." Journal of the American College of Cardiology 70.9 (2017): 1183-1192.

---

> ### Author Response · Authors · 2026-01-24
>
> We greatly appreciate your review. We took some of your comments/suggestions seriously, and addressed them underneath. Please refer to the following comments and revision.
>
> ---
>
> ### Validation paper track
>
> We appreciate this suggestion but, we still believe this paper would fit in the main track better than the validation track. We study how expert over-reads vs. machine-read interpretations affect learning and evaluation, enabled by a uniquely paired large-scale dataset. In contrast, validation track typically emphasizes rigorous external validation with multi-instutional data, which is challenging here because this paired triplet (machine-read, over-read from the same ECG) is not commonly available across sites. We explicitly acknowledge this as a limitation and discuss the need for multi-site validation as figure work.
>
> ---
>
> ### Supervised w/o augmentation
>
> Thank you for this great point. You are right that since self-training has augmentation in training, it might be fair for supervised training to need one for fair comparison. In the original submission, we intentionally used no-augmentation supervised baseline as a simple and commonly adopted baseline point. Regardless, we ran additional experiments with augmentation, and it improved the metrics. However, improvement was minimal, and still didn’t change the conclusion of the paper.
>
> ---
>
> ### Ablation study of available dataset sizes
>
> We agree that such ablation can be useful. We have already put this ablation study in our Appendix E. Sorry for the inconvenience, but we had to put this in the Appendix due to our page limit.
>
> ---
>
> ### Public code and model
>
> Yes, now our code and trained model weights are all available at
> https://github.com/tyoung089/Overread_ecg

---

### Official Review · Reviewer_ytF3 · 2026-01-17

**Confidence:** 3
**Preliminary Rating:** 4

**Summary:**

This paper studies how training with cardiologist over-reads effects model accuracy and clinical reliability. It uses large ECGs datasets containing both machine and expert interpretations, and evaluate different learning paradigms, accordingly. The methods used are 1- supervised learning on expert over-read labels, 2- Self-training that extends expert supervision to public ECGs, and 3-,4- Multimodal contrastive learning with CLIP and NegCLIP.

**Strengths:**

- Introducing a method that address misclassification of ECG automatic interpretation.
- Using large-scale dataset.
- Using and comparing four different approaches.
- A good and extensive comparison of the approaches used.
- Showing that expert over-reads are helpful annotations for building reliable ECG AI systems.

**Weaknesses:**

- The figures are trivial and should be improved to be more informative.
- More explanation of the methods used in the paper is required.
- The approaches used in this paper are based on previously developed methods and lack novelty.
- The paper would benefit from the inclusion of a conclusion section.

**Detailed Comments:**

- The figures should be made more professional, especially Figure 1.
- The parameter λ and its effect on the self-training method need to be explained.

**Justification Of The Preliminary Rating:**

- The paper has novelty in terms of application but not in methodology.
- The paper addresses an important issue in automatic ECG interpretation.
- Large-scale dataset is used to validate the methods.
- A good comparison between the methods used is provided.

**Questions To Address In The Rebuttal:**

- Explaining the parameter adjustments in a clearer and more detailed manner.
- Explaining where and how the method may perform poorly.

---

> ### Comment · Reviewer_btF2 · 2026-01-17
>
> I don't see in your review where you comment on the main stated contribution of the paper: "models trained
> with expert over-read data consistently outperform those trained on machine-read labels" which digging into the paper I understand is: comparing a model trained on automatically predicted labels (by inferior models) vs human predicted labels.
>
> I think this contribution of the paper is misleading as it sounds like something different than it is. "expert over-read labels" sounds like multiple experts reviewed the labels vs a single expert, but the experiments are not that.

---

> > ### Author Response · Authors · 2026-01-19
> >
> > Dear Reviewers ,
> >
> > First, we thank Reviewer btF2 for the comment. We would like to provide a brief clarification regarding the term over-read. Over-reading (or over-read) is a commonly used term in clinical practice and the medical literature, referring to cardiologists reviewing machine-generated ECG interpretations to ensure accurate interpretation and diagnosis (Smulyan, 2019; Schläpfer and Wellens, 2017). In our work, the term over-read does not refer to multi-expert or consensus readings. We will further clarify this point in the official rebuttal revision.
> >
> > We stated that “to address this challenge, we curated a large corpus of over two million 12-lead ECGs, each containing both the original machine-read interpretation and a cardiologist over-read provided by experts with more than 30 years of experience.” The over-read labels in our dataset were obtained through a standardized process in which experienced cardiologists systematically reviewed and corrected machine-read interpretations.
> >
> > We will clarify this description more explicitly in the upcoming revised rebuttal
> >
> > Thank you.
> >
> > Smulyan, Harold. "The computerized ECG: friend and foe." The American journal of medicine 132.2 (2019): 153-160.
> > Schläpfer, Jürg, and Hein J. Wellens. "Computer-interpreted electrocardiograms: benefits and limitations." Journal of the American College of Cardiology 70.9 (2017): 1183-1192.

---

> ### Author Response · Authors · 2026-01-24
>
> Dear Reviewer ytF3,
>
> Thank you very much for your constructive and thoughtful review of our work. We sincerely appreciate the time and effort you invested in providing detailed feedback. We have made our best effort to address all of your comments. Please note that some related points are addressed together below.
>
> ---
>
> ### Figures are trivial and should be more informative
>
> We agree with this assessment and have substantially improved the figures in the main text. We redesigned Figure 1 to more clearly and professionally illustrate the standardized clinical workflow for obtaining machine-read and expert over-read interpretations. We also added Figure 3 to the main text, replacing the prior ROC/PR curve visualization with a bar plot comparison that more directly contrasts independent diagnosis-statement text formulations with whole-report text formulations. We believe this new visualization makes performance differences easier to interpret at a glance.
>
> ---
>
> ### Explaining the method and parameters more clearly
>
> Thank you for pointing out this issue. We expanded the Method section to provide more detailed explanations of our approach and parameter choices, and added several sentences to clarify implementation details. In particular, the choice of the $\lambda$ parameter was determined through validation based tuning, and Appendix D was added to present empirical results supporting this choice in the self-training experiments.
>
> ---
>
> ### Inclusion of a conclusion section
>
> We fully agree with this suggestion. Since the revision allowed for an increased page limit, we added a dedicated Conclusion section at the end of the paper to formally summarize the work and its contributions.
>
> ---
>
> ### Where and how the method may perform poorly
>
> To address this concern, we expanded the limitations discussion at the end of the Discussion section. We explicitly note that our models are primarily trained and evaluated on data from a single institution, and that further validation using datasets from different sites is necessary to assess generalizability. We also discuss that our study is based on the assumption that expert over-read interpretations provide a stronger supervisory signal than machine-read outputs, an assumption that may evolve over time as vendor algorithms improve or vary across clinical settings.
>
> ---
>
> ### Lack of novelty
>
> We acknowledge that our learning framework builds upon established methods. However, we believe this work offers novelty in its application and empirical focus. In particular, we demonstrate that reliance on machine-read only supervision can be suboptimal, and that models trained using expert over-reads or explicitly emphasizing discrepancies between machine and expert interpretations consistently achieve better accuracy and precision. We believe these findings are clinically meaningful and contribute to the development of more reliable ECG AI systems.
>
> ---
>
> We again thank you for your insightful feedback, which helped improve the clarity and quality of this manuscript.

---

### Author Rebuttal · Authors · 2026-01-24

**Rebuttal:**

We greatly appreciate the reviewers for their valuable comments. We made our best effort to address all comments raised by each reviewer. Below, we provide a brief overview of the changes made in the revision. Detailed point by point responses to each reviewer are left under comments.

**Figures** -- ytF3 / btF2
- Redrew Figure 1 to more clearly illustrate how machine-read and expert over-read interpretations are obtained.
- Added Figure 3 to the main text to better compare the impact of text granularity on model performance.
- Updated captions of several figures and tables to improve clarity and descriptive accuracy.
- Added CI to the over all comparison results table.

**Methods** -- ytF3 / Wcvu
- Added additional details to the Method section, including clearer explanations of the approach and parameter choices.
- Added Appendix D to demonstrate that the choice of the λ parameter was guided by empirical results.
- Updated supervised results with augmentation.

**Discussion** -- ytF3 / btF2
- Expanded the Discussion section to better articulate the contributions of this work.
- Added a more thorough discussion of the study’s limitations.

**Conclusion** -- ytF3
- Added a Conclusion section to formally summarize the paper.

**Code Release** -- Wcvu / btF2
- Included a GitHub link to the public code and model weights at the end of the Abstract.

**Supporting Material:**

/attachment/028c009ef98f89ede05952b91c6012997e310935.pdf

---

### Author Response · Authors · 2026-01-26
**Concerns Regarding Reviewer btF2's Assessment**

Dear Program Chairs, and Area Chairs,

We respectfully bring to your attention concerns regarding Reviewer btF2's assessment.

**Formatting-focused weaknesses**: The majority of the reviewer's criticisms concern presentation issues—appendix reliance, figure captions, and figure formatting. These constraints arise from the page limit and can be readily addressed with the extended page allowance in the camera-ready version.

**Mischaracterization of our claims**: The reviewer states that "over-read labels are multiple expert reviewed labels," then critiques our motivation based on this interpretation. However, we did not make this claim in our paper. More concerningly, the reviewer attempted to communicate this mischaracterization to other reviewers before our rebuttal and discussion period which may have influenced their judgement.

We kindly ask the AC and PC to consider these factors when evaluating the final recommendation.

Thank you for your careful consideration.

---

### Meta-Review · Area_Chair_eEoE · 2026-02-03

**Recommendation:** Accept (Poster)
**Confidence:** 5

**Metareview:**

This work investigates the value of incorporating cardiologist over-read interpretations to improve ECG classification accuracy. The study relies on several datasets, most notably a large internal dataset comprising over two million ECGs with both machine-generated and expert interpretations, which is exceptional. As noted by several reviewers, although the paper does not present strong intrinsic technical novelty, it provides meaningful analyses and the experiments are conducted very thoroughly on large-scale datasets.

The discussion and rebuttal phases were handled seriously by the authors, leading to a clear improvement in the quality of the manuscript, notably through the revision of key figures, the inclusion of confidence intervals in table 1, and the release of public code and pre-trained model weights via a GitHub repository. For these reasons, I recommend acceptance of this paper.

---

### Decision · Program_Chairs · 2026-02-13

Accept (Poster)